# Study on Water Rights Allocation of Irrigation Water Users in Irrigation Districts of the Yellow River Basin

Xinjian Guan [1], Baoyong Wang [1], Wenge Zhang [2,3,*] and Qiongying Du [1]

1   School of Water Resources Science and Engineering, University of Zhengzhou, Zhengzhou 450001, China; gxj1016@zzu.edu.cn (X.G.); wangbaoyong2020@163.com (B.W.); dqy13783451709@163.com (Q.D.)
2   Yellow River Institute of Hydraulic Research, Yellow River Conservancy Commission, Zhengzhou 450003, China
3   Henan Key Laboratory of Ecological Environment Protection and Restoration of Yellow River Basin, Zhengzhou 450003, China
*   Correspondence: zhangwenge@yeah.net; Tel.: +86-135-2684-0719

**Abstract:** With the increasingly serious problems of water security and water shortage in the Yellow River Basin, the establishment of a fair and efficient water rights distribution system is an important way to improve water resource utilization efficiency and achieve high-quality development. In this paper, a double-level water rights allocation model of national canals–farmer households in irrigation districts is established. The Gini coefficient method is used to construct the water rights allocation model among farmer households based on the principle of fairness. Finally, the Wulanbuhe Irrigation Area in the Hetao Irrigation District is taken as an example. Results show that the allocated water rights of the national canals in the irrigation district are less than the current; for example, water rights of the Grazing team (4) canal are reduced by 73,000 m$^3$ than before, in which water rights of farmer households 1, 2, 3, and 4 obtain compensation and 5, 6, 7, and 8 are cut by the water rights allocation model and the Gini coefficient is reduced from 0.1968 to 0.1289. The research has fully tapped the water-saving potential of irrigation districts, improved the fairness of initial water rights distribution, and can provide a scientific basis for the development of water rights allocation of irrigation water users in irrigation districts of the Yellow River Basin.

**Keywords:** Gini coefficient; fairness principle; double-level; water-saving potential

## 1. Introduction

Agriculture accounts for 70% of global water withdrawals, most of which is used for irrigation, so it is particularly important to carry out research on the distribution of agricultural water rights in irrigation areas to alleviate the current water shortage problems [1]. The initial allocation of water rights is the first step in the construction of a water rights system and the key measure to carry out water rights trade and give play to the function of optimal allocation of market resources. Based on the experience at home and abroad, the modern water rights system can be divided into the riparian rights system, the priority occupancy rights system, and the public water distribution rights system according to the initial acquisition and distribution forms of water rights [2]. Currently, China implements an administration-led public water rights allocation system [3]. The distribution system is generally from top to bottom, which distributes the initial water rights in a basin to provinces, cities, counties, industries, and final water users [4].

In recent decades, so many scholars have conducted a lot of research on initial water rights distribution, and early research mainly distributes initial water rights from the perspective of fairness [5,6], comprehensively considering the land area, capital investment, public law, water priority, water licenses, and reasonable collection of water fees, etc. [7–9], which enrich the insufficient system of the authorization and water permission system in the original irrigation area. With the in-depth study, some research on initial water

rights distribution technology has also been carried out. Based on the conditional value at risk theory and Gini coefficient constraints, Zhang L.N. [10] establishes a two-stage stochastic programming model for water rights distribution, which reduces the unfair risk of local water shortages. Sahebzadeh Ali [11] uses the concept of conditional value at risk (CVaR) in the water distribution model to minimize the water loss index under low flow conditions. Using the automatic biophysical surface energy balance model (BAITSSS), Ramesh Dhungel [12] studies two agriculturally dominated groundwater areas in the northwest of the United States and the irrigation simulated by the model is compared with the report on the water rights management unit (WRMU). Imron F [13] uses linear programming to analyze the optimization of irrigation water distribution. By combining the water evaluation and planning system model and the non-principal sorting genetic algorithm II (NSGA-II) optimization algorithm, Chakraei Iman [14] puts forward a comprehensive simulation optimization model for the Zayanderud River Basin in Iran, and the distribution of surface and groundwater resources to various agricultural regions is optimized. Gebre Sintayehu Legesse [15] studies the application of multi criteria decision making (MCDM) related to water resource allocation. In addition, some scholars consider climate change, reservoir operation capacity, regional economic development, and other factors to establish a multi-objective optimization model to realize the fair distribution of water [16–18].

At present, the initial distribution of water rights is mainly concentrated on the distribution from a basin to regions and industries. It is a multi-objective and multi-level distribution problem that the water rights obtained by provinces are further allocated to cities and counties. When the superior water rights allocation method is applied to the county level, there are problems such as large differences in water use among towns, inapplicability of the allocation index system, and difficulty in collecting specific data and so on [19]. The second layer of allocation of water rights is subject to the principle of priority under the constraint of total control among industries to construct a target planning model based on the principles of priority of domestic water, food security, attention to ecological environment, economic benefits, and reasonable industrial structure [20]. Therefore, in the process of initial water rights distribution in the irrigation area, it is an inevitable requirement to further allocate the irrigation water rights to the main body of irrigation water users to realize the refinement of agricultural water management. The existing agricultural water distribution system mostly takes the irrigation area as the minimum distribution unit.

In this paper, according to the characteristics of multi-level water consumption in irrigation districts, a double-level water rights allocation model of national canals–farmer households in the irrigation district is established. The total amount of water rights distribution in national canals is determined by considering the future water-saving potential of the irrigation area. At the farmer household level, the fairness of water rights distribution is fully considered in combination with the characteristics of asymmetric information of farmers' agricultural population and irrigation area. Finally, the Wulanbuhe Irrigation Area of Hetao Irrigation District in the Yellow River Basin is taken as an example for verification based on the double-level water rights allocation model, and the research results can provide new ideas and methods for regional unit agricultural water rights allocation.

## 2. Materials and Methods

### 2.1. Overview of the Study Area and Data Sources

2.1.1. Overview of the Study Area

Wulanbuhe Irrigation Area is located in the west of the Hetao Irrigation District of Inner Mongolia, and it mainly involves three administrative districts of Dengkou, Hangjin Houqi, and Azuo Qi. The total population of the irrigation area is 115,100, including a rural population of 69,100, and the irrigation area is 68,100 hm$^2$ in 2017. Wulanbuhe Irrigation Area belongs to the inland high plain of Hetao basin, located in the northeast of Wulanbuhe Desert. It belongs to the temperate continental monsoon climate, with four

distinct seasons, abundant sunlight, large temperature differences, and rare precipitation. The average annual precipitation is 144.5 mm, and the average annual evaporation is 2377.1 mm. The local water resources are very scarce. In order to meet the local water demand, it is necessary to use the transit Yellow River water, which has a certain water intake index for this area. Wulanbuhe Irrigation Area depends mainly on the Yellow River water for irrigation by the Shenwu main canal; there are a total of 476 main canals and sub-main canals in the Wulanbuhe Irrigation Area, of which 411 canals diverted directly from national canals are confirmed, because the water rights of the 411 canals will be distributed directly to the corresponding farmers, and so this article focuses on the distribution of the Yellow River water rights for those canals in the irrigation district. The basic situation of Wulanbuhe Irrigation Area is shown in Figure 1.

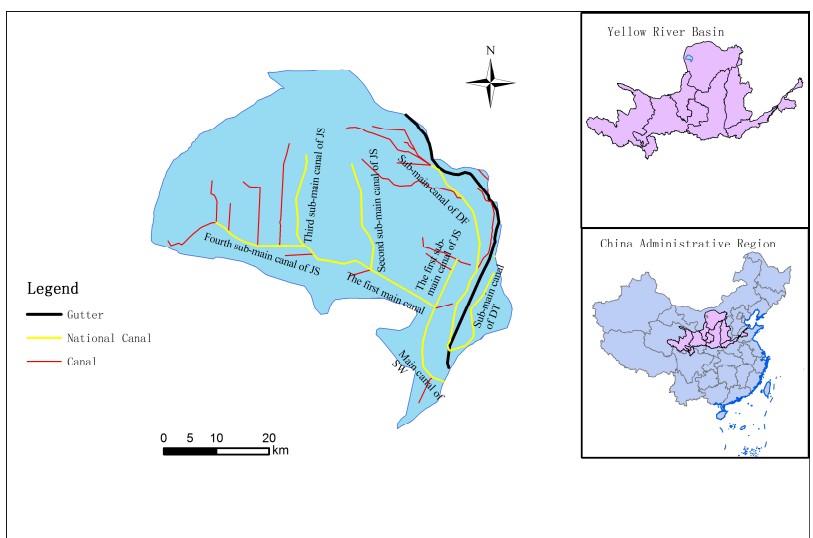

**Figure 1.** Wulanbuhe Irrigation Area of Hetao Irrigation District.

2.1.2. The Data Source

There are 411 canals diverted directly from national canals that are confirmed in the Wulanbuhe Irrigation Area of Hetao Irrigation District. The administration of the Hetao Irrigation District has made statistics for the five-year water consumption of these canals in the Wulanbuhe Irrigation Area from 2008 to 2013 (excluding 2012 due to a larger water shortage than usual), and the data are true. According to the proposed plan of water-saving irrigation engineering, the water-saving volume of the irrigation fields in the future can be calculated. The population and irrigation area of the corresponding farmer households in these canals were obtained from the actual statistical results of the township.

*2.2. Double-Level Water Rights Allocation Model of the Irrigation District*

The double-level water rights allocation model for the irrigation district includes the distribution method of water rights at the level of the national canal system and the distribution method of water rights among farmer households. Taking the amount of water diversion from the main canal head of the irrigation district as the total amount of water rights allocation, firstly allocate water rights at the national canal system level, and then use those as the total for water rights allocation among farmers. The canal system structure diagram of the irrigation district is shown in Figure 2.

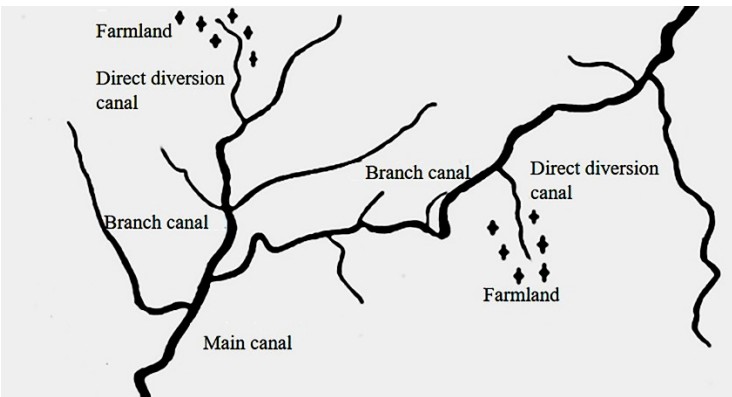

**Figure 2.** National canal system structure diagram of irrigation district.

### 2.2.1. Water Rights Allocation Model of National Canal System in Irrigation District

(1) Total amount of current water rights at the national canal system level. Generally, the total amount of canal system water rights is determined by the actual water diversion in the irrigation district and the average water consumption over the years.

(2) Water-saving potential of the irrigation district. The main water-saving measures in the irrigation district are canal lining, border field reconstruction, and drip irrigation. The total water-saving amounts of water-saving projects in the irrigation district is the canal-level water-saving amount. The calculation formula is as follows:

Water-saving amount of canal system:

$$\Delta W_i = \Delta W_{ic} + \Delta W_{iq} + \Delta W_{id} \tag{1}$$

Water-saving amount of canal lining:

$$\Delta W_{ic} = W_i(1 - \eta_i) - W'_i(1 - \eta'_i) \tag{2}$$

Water-saving amount in border field reconstruction:

$$\Delta W_{i\,q} = W_{iqb} - W_{iql} \tag{3}$$

Water-saving amount of drip irrigation:

$$\Delta W_{id} = W_{idb} - W_{idl} \tag{4}$$

where $\Delta W_i$ is the water-saving amount of the canal $i$, m$^3$; $\Delta W_{ic}$, $\Delta W_{iq}$, $\Delta W_{id}$ are, respectively, the water-saving amounts of canal lining, border field reconstruction, and drip irrigation of the canal $i$, m$^3$; $W_i$, $W'_i$ are, respectively, the canal head water intakes before and after the lining of the canal $i$, m$^3$; $\eta_i$, $\eta'_i$ are, respectively, the canal system water utilization coefficients before and after the lining of the canal $i$; ($0 < \eta_i < \eta'_i < 1$); $W_{iqb}$, $W_{iql}$ are the field irrigation amounts before and after the renovation of border fields of the canal $i$, m$^3$; $W_{idb}$, $W_{idl}$ are the headwater diversions before and after drip irrigation reconstruction of the canal $i$, m$^3$.

(3) Distribution of water rights of national canal system. By analyzing the total amount of current water rights of the canal system in the irrigation district and considering the potential water-saving amount of the canal system in the future, the canal-level water rights allocation model is determined. The calculation formula is as follows:

$$W_{ip} = W_{is} - \Delta W_i \tag{5}$$

where $W_{ip}$ is the water rights distribution of the canal $i$, m$^3$; $W_{is}$ is the total amount of current water rights of the canal $i$, m$^3$.

Due to the constraint of the water diversion permit in irrigation districts, the total amount of water rights allocated at the canal level shall not exceed the permitted amount. Under the constraint of the water intake permit, canal-level water rights allocation in irrigation districts is as follows:

When the allowable water intake is more than the actual total water diversion of each canal directly from national canal, that is:

$$\sum_{i=1}^{n} W_{ip} \leq W_Q \tag{6}$$

$$W_{ip} = W_{is} - \Delta W_i \tag{7}$$

When the allowable water intake is less than the actual total water diversion of each canal directly from national canal, that is:

$$\sum_{i=1}^{n} W_{ip} \geq W_Q \tag{8}$$

$$W_{ip} = \lambda_{ip} \times W_Q \tag{9}$$

$$\lambda_{ip} = \frac{W_{ip}}{\sum\limits_{i=1}^{n} W_{ip}} \tag{10}$$

where $W_Q$ is the allowance of water intake in the irrigation district, m$^3$; $\lambda_{ip}$ is the water distribution coefficient of the canal $i$.

2.2.2. Water Rights Allocation Model among Farmer Households in Irrigation Districts

(1)  Select the indexes of water rights allocation among farmer households

   (i)  Irrigation area of farmer households

Current agricultural water rights allocation is based on irrigation area. The larger the irrigation area, the more water rights are allocated. The distribution of water rights according to the irrigation area mainly reflects the difference of irrigation water of different farmer households, and the distribution of water rights according to irrigation area is as follows:

$$S_j = q \times a_j \tag{11}$$

$$q = \frac{W_{ip}}{A_i} \tag{12}$$

where $S_j$ is the water rights of farmer household $j$ distributed, m$^3$; $q$ is the water rights allocation quota, m$^3$/hm$^2$; $a_j$ is the irrigation area of farmer household $j$, hm$^2$; $A_i$ is the irrigation area confirmed for all farmers in the canal system, hm$^2$; $W_{ip}$ is the water rights distributed of the canal $i$, m$^3$.

   (ii)  Peasant household agricultural population

Water resources are the public resources of the whole society, so the distribution of water rights should give consideration to the development of all people, and the agricultural population of peasant households should be fully considered in the distribution of water rights. The household with more (less) agricultural population will obtain more (less) water rights. The distribution process is as follows:

$$S_j = q \times p_j \tag{13}$$

$$q = \frac{W_{ip}}{P_i} \tag{14}$$

where $S_j$ is the water rights distributed for farmer household $j$, m$^3$; $q$ is the water rights allocation quota, m$^3$/hm$^2$; $p_j$ is the agricultural population of farmer household $j$; $P_i$ is the agricultural population of all farmer households of the canal $i$; $W_{ip}$ is the water rights distributed for the canal $i$, m$^3$.

(2) Water rights allocation model among farmer households based on Gini coefficient method

(i) Gini coefficient

The Gini coefficient [21], also known as the Lorentz coefficient, was first proposed by Italian mathematician Gini at the beginning of the 20th century. It is mainly used in the field of economics to investigate and measure the inequality of regional residents' income and wealth distribution. It can more directly reflect the income difference between residents.

The value range of the Gini coefficient is [0, 1]. When the Gini coefficient is 0, it represents the absolute average of income distribution. Moreover, 0.4 is usually regarded as the warning line of the income gap in the world, and the evaluation standard of the Gini coefficient can be referred to the following Table 1.

**Table 1.** Gini coefficient evaluation criteria.

| Gini Coefficient | <0.2 | 0.2~0.3 | 0.3~0.4 | 0.4~0.5 | >0.5 |
|---|---|---|---|---|---|
| Evaluation results | Absolute average | Comparative average | Relatively reasonable | Big gap | Wide disparity |

(ii) Construction of water rights allocation model by Gini coefficient method

When a peasant household's water rights are distributed based on irrigation area and the farmer household's agricultural population are equal, the water rights allocation is considered to be fair. When the water rights allocated are not same, neither of the two distribution patterns can reflect the principle of fairness in the allocation of water rights; meanwhile, the irrigation area of farmer households and the agricultural population of farmer households are asymmetrical. In this article, therefore, the per capita irrigation area of each farmer is used as a measure of the fairness of water rights allocation, and the theory of the Gini coefficient is used to study the distribution relationship between irrigation area of farmer households and their agricultural population. With the cumulative percentage of the agricultural population of each farmer household in the canal system as the abscissa and the cumulative percentage of the irrigated area of each farmer household as the ordinate, the water rights allocation model was built based on minimizing the Gini coefficient. The specific steps are as follows:

Step 1: Building the objective function

$$\min G_{ini} \tag{15}$$

$$G_{ini} = \frac{A}{A+B} = 2A = 1 - 2B = 1 - \sum_{j=1}^{n}(X_j - X_{j-1})(Y_j + Y_{j-1}) \tag{16}$$

$$(Y_j - Y_{j-1})A_i = x_j \times (X_j - X_{j-1}) \times P_i \tag{17}$$

where $X_j$ is the cumulative percentage of agricultural population of farmer household $j$; $Y_j$ is the cumulative percentage of irrigation area after equilibrium of farmer household $j$; $P_i$ is the corresponding total agricultural population of the canal $i$; $A_i$ is the corresponding total irrigation area of the canal $i$, hm$^2$; $x_j$ is the per capita irrigation area after equilibrium of farmer household $j$, hm$^2$/person.

The Lorenz curve of the population and irrigation area is as follows in Figure 3.

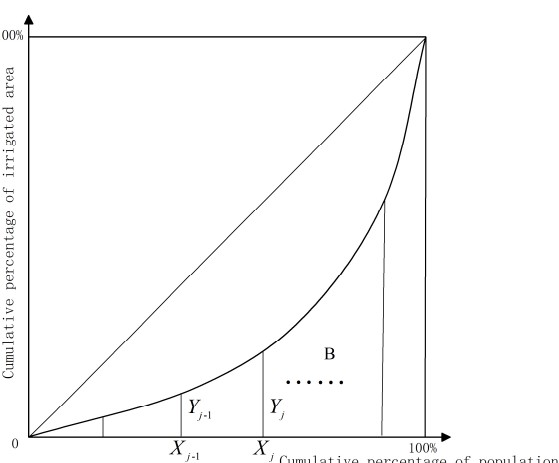

**Figure 3.** Population–irrigation area Lorentz curve.

Step 2: Setting constraints:
a: Fairness constraints:

$$\begin{cases} x_j > x'_j \, , & x_j < \overline{x}_i \\ x_j < x'_j \, , & x_j > \overline{x}_i \end{cases} \tag{18}$$

where $x_j$ is the per capita irrigation area after equilibrium of farmer household $j$, $hm^2$/person; $x'_j$ is the current per capita irrigation area of farmer household $j$, $hm^2$/person; $\overline{x}_i$ is the per capita irrigation area of farmers of the canal $i$ $hm^2$/person.

b: Constraints of basic water security:

$$\left| \frac{x'_j - x_j}{x'_j} \right| \le s \tag{19}$$

where $s$ is the reduction ratio determined by the degree of importance the region attaches to the principle of equity.

Restrictions on the extent of reduction or compensation:

$$\left| x_j - x'_j \right| \ge \left| x_\rho - x'_\rho \right|, \qquad \left| x'_j - \overline{x}_i \right| \ge \left| x'_\rho - \overline{x}_i \right| \, , \, j \ne \rho \tag{20}$$

c: Constraints of sorting:

$$x_{j-1} \le x_j \le x_{j+1} \tag{21}$$

d: Constraints on irrigation area:

$$\sum_{j=1}^{n} x_j \times p_j = A_i \tag{22}$$

where $p_j$ is the agricultural population of farmer household $j$; $A_i$ is the corresponding total irrigation area of the canal $i$, $hm^2$.

e: The Gini coefficient after equilibrium is smaller than before:

$$G_{ini} < G'_{ini} ini \tag{23}$$

f: Non-negative constraints:

$$x_j > 0 \tag{24}$$

Step 3: Determining the water rights of farmers distributed:

$$W_{jp} = \frac{W_{ip}}{\sum\limits_{j=1}^{n} x_j \times p_j} \times x_j \times p_j \tag{25}$$

where $W_{jp}$ is the water rights of farmer j distributed, m$^3$; other symbols are the same as above.

Step 4: Optimal solution of the model:

The model is optimized and solved by the genetic algorithm in MATLAB, and the calculation process of the optimized solution is as follows:

a: At the beginning of the genetic algorithm calculation, first set various parameters, such as setting the population size to 20, the number of iterations, the probability of crossover and mutation, and the termination conditions.

b: Generate the initial value group for the per capita irrigation area of farmers: $pop = [z_1, z_2, z_3, z_4, z_5, z_6, z_7, z_8]$.

Define fitness function: $G_{ini} = \left[1 - \sum\limits_{j=1}^{n} (X_j - X_{j-1})(Y_j + Y_{j-1})\right]$ and then calculate the fitness of the initial population and compare the fitness value of the population.

c: Set the constraint conditions to see whether the fitness of the initial population meets the optimization criterion. If it is satisfied, the optimization ends; if not, proceed to step d.

d: Select, cross, and mutate on the initial population $pop$, to produce offspring population $pop_1$, and see whether the population $pop_1$ meets the optimization conditions. If it is satisfied, the optimization ends; if it is not satisfied, the selection, crossover, and mutation operations are continued until the conditions are met.

The optimization flowchart is as follows in Figure 4, and we complete this part based on MATLAB 2018B.

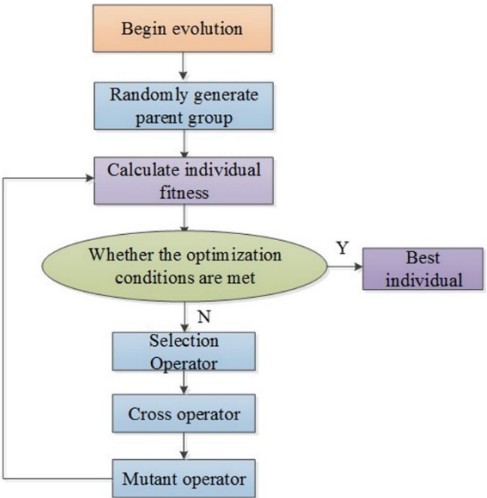

**Figure 4.** Genetic algorithm optimization flowchart.

## 3. Results

### 3.1. Distribution Results of Water Rights for the Canals Diverted Directly from the National Canal System

To allocate canal-level water rights for 411 canals diverted directly from the national canal system that need to be confirmed in the Wulanbuhe Irrigation Area, according to the current situation of water-saving projects in the Wulanbuhe Irrigation Area, Formulas (1)–(4) are used to calculate the water-saving amount of the 411 canals. In addition, based on the five-year average water volume collected for the canals diverted directly

from national canal system, Formula (5) and Formulas (6)–(10) are adopted to calculate the distribution of water rights for the 411 canals. Take one of the 411 canals in the Wulanbuhe Irrigation Area as an example for explanation, as shown in Table 2.

**Table 2.** Results of water rights distribution for canals diverted directly from the national canal system in Wulanbuhe Irrigation Area (ten thousand m³).

| Direct Diversion Canal Name | Township (Farm) | Five-Year Average Water Volume | Water Saving | | | Water Rights Allocation |
| --- | --- | --- | --- | --- | --- | --- |
| | | | Water Saving in Canal Lining | Water Saving in Border Field Reconstruction | Water Saving in Drip Irrigation | |
| Grazing team (4) | Bayangaole Town | 20.3 | 0 | 7.3 | 0 | 13.0 |
| Bayi canal | Wulanbuhe Farm | 1981.5 | 0 | 421.6 | 0 | 1559.8 |
| The fourth lateral canal | Hatengtaohai Farm | 843.5 | 229.6 | 151.4 | 0 | 462.5 |
| New third canal | Bayantauhai Farm | 169.5 | 61.0 | 80.8 | 0 | 27.6 |
| First canal of four groups | Sun Temple Farm | 336.7 | 0 | 264.4 | 0 | 72.3 |
| Susan canal 1 | Shajin Sumu | 137.4 | 0 | 61.7 | 0 | 75.7 |
| Two rounds of water 1 | Experiment Bureau | 354.2 | 0 | 122.4 | 0 | 231.8 |
| Western third lateral canal | Narintaohai Farm | 672.8 | 0 | 260.3 | 0 | 512.5 |
| The fourth brunch canal | Baoergai Farm | 5633.0 | 1463.2 | 0 | 2009.6 | 2160.2 |
| Zhao Duozhi | Bayin Maodao Gacha | 126.6 | 88.2 | 15.7 | 0 | 22.7 |
| Loess file one | Bulongnao Town | 213.9 | 0 | 0 | 36.8 | 177.1 |
| Tuanjie branch canal | San Tuan Farm | 3049.8 | 808.3 | 0 | 328.8 | 1912.7 |
| First lateral canal | Longsheng Hezhen | 365.1 | 0 | 70.0 | 0 | 295.1 |

### 3.2. Results of the Water Rights Distribution among Farmer Households

Since the patterns of water rights allocation among farmer households for 411 canals are the same, the canal of Grazing team (4) in Bayangaole Town is taken as an example for calculation and analysis. The irrigation area and the agricultural population of farmer households are selected as the water rights allocation indexes under asymmetric information. On the basis of the calculation formula of the Gini coefficient, the population of farmer households and the corresponding irrigation area data are arranged according to the per capita irrigation area from small to large. The calculation process is shown in Table 3.

**Table 3.** The relevant calculation results for the Gini coefficient under the current condition.

| Farmer Household Number | Irrigation Area (hm²) | Agricultural Population | Current per Capita Irrigation Area (hm²/Person) | $(X_j - X_{j-1}) * (Y_j + Y_{j-1})$ |
| --- | --- | --- | --- | --- |
| 1 | 0.333 | 6 | 0.056 | 0.0107 |
| 2 | 0.533 | 8 | 0.067 | 0.0515 |
| 3 | 0.400 | 5 | 0.080 | 0.0572 |
| 4 | 0.333 | 4 | 0.083 | 0.0615 |
| 5 | 0.733 | 6 | 0.122 | 0.1267 |
| 6 | 0.933 | 7 | 0.133 | 0.2104 |
| 7 | 0.600 | 4 | 0.150 | 0.1531 |
| 8 | 0.467 | 3 | 0.156 | 0.1320 |
| total | 4.333 | 43 | | 0.8032 |

According to the above calculation results, the Gini coefficient for the current distribution of water rights is 0.1968. The above data are substituted into the water rights allocation model among farmer households, and then the balanced per capita irrigation area for eight farmer households of Grazing team (4) are determined through objective function Equations (15)–(17) and constraint Equations (18)–(24).

The current per capita irrigation area of farmer households which exceeds (falls short of) the average per capita irrigation area of the canal system, =0.101 hm$^2$/person, needs to be reduced (compensate). The fairness constraint is:

$$\begin{cases} x_1 \geq 0.056 \\ x_2 \geq 0.067 \\ x_3 \geq 0.080 \\ x_4 \geq 0.083 \\ x_5 \leq 0.122 \\ x_6 \leq 0.133 \\ x_7 \leq 0.150 \\ x_8 \leq 0.156 \end{cases}$$

A substantial reduction in the per capita irrigation area of farmer households will lead to a reduction in their allocated water rights. In order to ensure a certain amount of basic irrigation water for farmers, we consulted the local water resources management department. This paper sets the reduction ratio to 0.3, and the basic water security constraint is:

$$\begin{cases} x_5 \geq 0.085 \\ x_6 \geq 0.093 \\ x_7 \geq 0.105 \\ x_8 \geq 0.109 \end{cases}$$

The more per capita irrigation area is above or below the average of canal system, $\bar{x} = 0.101$ hm$^2$/person, the greater the degree of reduction or compensation is; that is, the degree of reduction and compensation is restricted as follows:

$$|x_8 - 0.156| \geq |x_7 - 0.150| \geq |x_1 - 0.056| \geq |x_2 - 0.067| \geq |x_6 - 0.133| \geq |x_5 - 0.122| \geq |x_3 - 0.080| \geq |x_4 - 0.083|$$

After the equilibrium, the per capita irrigation area of each farmer household still satisfies the ranking before the equilibrium, ensuring the fairness of the distribution of water rights among farmer households; that is, the ranking constraint is:

$$6x_1 + 8x_2 + 5x_3 + 4x_4 + 6x_5 + 7x_6 + 4x_7 + 3x_8 = 4.333$$

After the equilibrium, the Gini coefficient of the farmer households' agricultural population–irrigation area should be smaller than that before the equilibrium, to ensure that the distribution plan is fairer than the current distribution; that is:

$$G_{ini} \leq 0.1968$$

After equilibrium, the per capita irrigation area of each farmer household is greater than 0; that is, the non-negative constraint is:

$$x_i \geq 0 \quad (i = 1, 2, \ldots, 8)$$

The genetic algorithm in MATLAB is used to solve the model, and the per capita irrigation area of the eight farmer households of the canal of Grazing team (4) is balanced. The balanced per capita irrigation area of the farmer households is shown in Table 4.

**Table 4.** The per capita irrigation area of each farmer household after equilibrium.

| Farmer Household Number | Area (hm$^2$) | Agricultural Population | Per Capita Irrigation Area of Farmer Household (hm$^2$/Person) | Per Capita Irrigation Area of Farmer Household after Equilibrium (hm$^2$/Person) |
|---|---|---|---|---|
| 1 | 0.333 | 6 | 0.056 | 0.071 |
| 2 | 0.533 | 8 | 0.067 | 0.079 |
| 3 | 0.400 | 5 | 0.080 | 0.087 |
| 4 | 0.333 | 4 | 0.083 | 0.089 |
| 5 | 0.733 | 6 | 0.122 | 0.115 |
| 6 | 0.933 | 7 | 0.133 | 0.122 |
| 7 | 0.600 | 4 | 0.150 | 0.133 |
| 8 | 0.467 | 3 | 0.156 | 0.137 |
| Total | 4.333 | 43 | | |

According to the canal-level water rights allocation method, the allocated water rights of the canal of Grazing team (4) is 13,000 m$^3$. According to the per capita irrigation area of each farmer household after the equilibrium, combined with Formula (25), the water rights distributed for each farmer household by the model is calculated. The current per capita irrigated area and the per capita irrigated area after equilibrium are shown in Figure 5 and the current water rights of farmer households and the water rights distributed by the model are shown in Table 5.

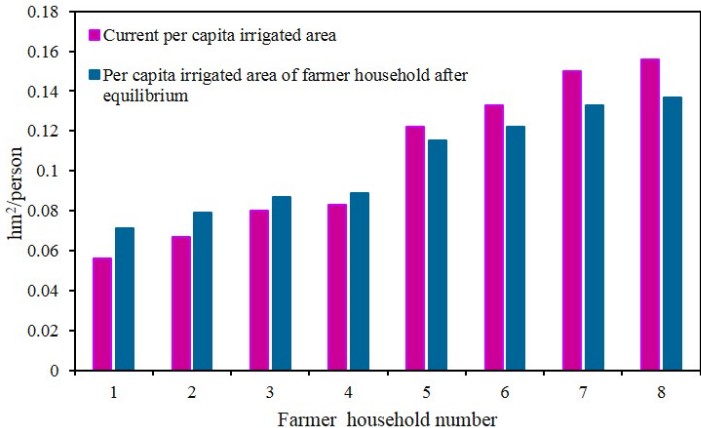

**Figure 5.** Per capita irrigation area of farmer households before and after equilibrium.

**Table 5.** The amount of water rights allocated by the model and the amount of current allocated water rights for farmer households (m$^3$).

| Farmer Household | 1 | 2 | 3 | 4 | 5 | 6 | 7 | 8 |
|---|---|---|---|---|---|---|---|---|
| Water rights allocated by the model | 1284 | 1888 | 1310 | 1072 | 2064 | 2562 | 1600 | 1220 |
| Current allocated water rights | 1000 | 1600 | 1200 | 1000 | 2200 | 2800 | 1800 | 1400 |

## 4. Discussion

### 4.1. Analysis of Water Rights Distribution for the Canals Diverted Directly from the National Canal System

The current actual water consumption compared with the permitted water volume shows that the total water consumption volume of the canal system in the Wulanbuhe Irrigation Area is 347.9529 million m$^3$, which is greater than the 330 million m$^3$ permitted. The current water rights allocation needs to be adjusted.

After the completion of the water-saving project, the total amount of water rights distribution for the canal system can be reduced. The Wulanbuhe Irrigation Area mainly saves water through three water-saving projects of canal lining, border field reconstruction, and drip irrigation. The future water-saving amount calculated of three water-saving projects is 77.641 million m$^3$, 68.10 million m$^3$, and 22.40 million m$^3$, respectively. The total water saving in the Wulanbuhe Irrigation Area is 168.0817 million m$^3$.

According to the water rights allocation model of the national canal system, from the actual current water volume minus the water-saving amount, the total amount of water rights allocated to the 411 canals is 179.8712 million m$^3$, which is less than the permitted water volume, and there is a remaining water volume of 150.1288 million m$^3$. The remaining water can be traded for water rights to increase the efficiency of water resources utilization. The relevant water volume is shown in Figure 6.

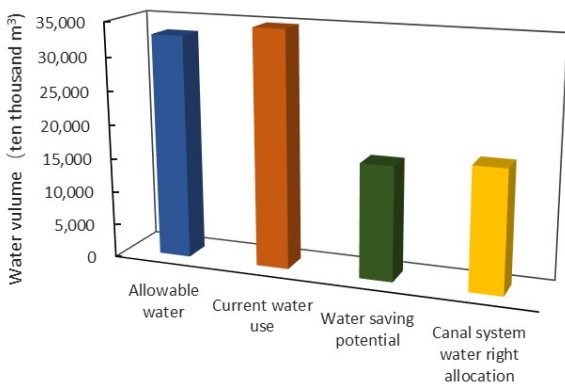

**Figure 6.** Water rights allocation of canal system in Wulanbuhe Irrigation Area.

*4.2. Performance Test of Water Rights Allocation Model among Farmer Households*

After the optimization of the model is solved, the Gini coefficient of the farmer household's population–the balanced irrigation area of the Grazing team (4) is 0.1289, which has been significantly improved compared with the Gini coefficient of 0.1968 of the farmer household's population–the current irrigation area, and the distribution of water rights among farmer households through the model is more equitable. Comparing the per capita irrigation area of farmer households after equilibrium by the model with that of before, the compensation for farmers 1, 2, 3, and 4 is 0.0158 hm$^2$/person, 0.0120 hm$^2$/person, 0.0073 hm$^2$/person, and 0.006 hm$^2$/person, respectively, and the reduction for farmers 5, 6, 7, and 8 is 0.0075 hm$^2$/person, 0.0113 hm$^2$/person, 0.0167 hm$^2$/person, and 0.0189 hm$^2$/person, respectively. The water rights distributed by the model for each farmer household have also been compensated or reduced accordingly, compared to before. The amount of compensation (reduction) is shown in Figure 7.

According to the distribution results of the model, for farmer households with a small population and large irrigation area, such as farmer households 5, 6, 7, and 8, the water rights allocated by the model are less than the current allocation. As their irrigation needs cannot be met, they can adjust planting structures or obtain additional water rights through water rights transactions. For farmer households with a large population and a small irrigation area, such as farmer households 1, 2, 3, and 4, the water rights allocated by the model are 754 m$^3$ more than the current allocation when only the irrigation area is considered. The allocation results by the model take into account the asymmetric factors of farm household population and irrigation area, and is more equitable. For example, the current water rights distributed for farmer household 1 and farmer household 4 are both 1000 m$^3$, but the water rights allocated by the model are 1284 m$^3$ and 1072 m$^3$, respectively. This is precisely considering the factor of farmer household population, where relatively more water rights are allocated for farmer households with larger populations.

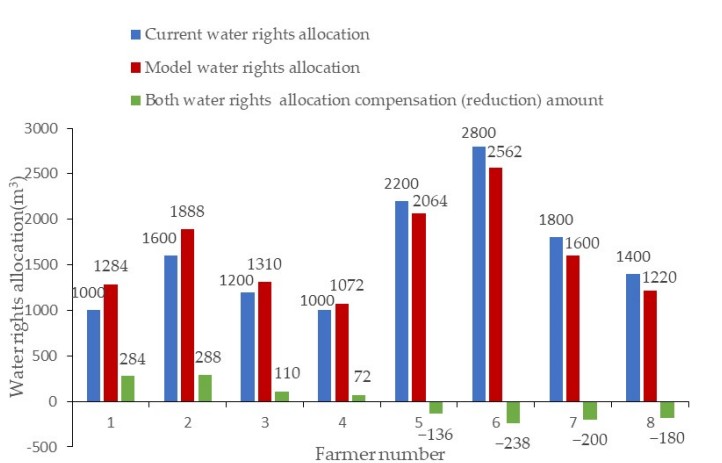

**Figure 7.** Comparison chart of current water rights and water rights allocated by the model.

*4.3. Overall Analysis of Water Rights Distribution in the Irrigation District*

The results of the water rights distribution at the national canal system level and among farmer households calculated by the double-level water rights allocation model show that the total amount of water rights allocated for each canal in the irrigation district has been greatly reduced, which will inevitably lead to a relative decrease in the water rights distributed for farmers in the irrigation district. The distribution of water rights in irrigation areas needs to comprehensively consider fairness and efficiency, but most of the existing studies only consider the area of agricultural land and the actual irrigation area of agricultural land [22], with a lack of consideration for the asymmetry between farmers' population and irrigation area [23,24]. In order to ensure the fairness of agricultural water rights distribution, it is necessary to comprehensively consider the agricultural population and irrigation area in the irrigation water user water rights distribution system. The water rights distribution model among farmers established in this paper is more fair in the process of water rights distribution, and alleviates the contradiction between farmers and water distribution managers to a certain extent. After the establishment of a farmers' water rights market, farmers with more water rights voluntarily sell water rights, while farmers with less water rights actively purchase water rights, which provides an opportunity for water rights trading among farmers in irrigation areas.

**5. Conclusions**

The rational distribution of agricultural water rights in irrigation areas is an important basis for improving the agricultural water rights system and establishing a water rights market. This paper establishes a double-level water rights allocation model of canals–farmers in an irrigation district, which is applied to the water rights distribution of the Wulanbuhe Irrigation Area in the Yellow River Basin. The main conclusions are as follows:

(1) Combined with the future water-saving potential of the canal system control area in the irrigation area, the canal system level water rights distribution model is established. Considering the factors of farmers' agricultural population and irrigation area, the water rights distribution model at the farmers' level based on the Gini coefficient method is established, which compensates the water users whose per capita irrigation area is less than that of the canal system, and fully reflects the fairness and enriches the existing theoretical system of initial water rights allocation.

(2) The government should strengthen the investment in water-saving projects, promote efficient irrigation technology, and fully tap the water-saving potential. Farmers should pay attention to the implementation of field water-saving measures, adjust the planting structure, and actively respond to the government's call to improve their self-awareness of water-saving. Realizing the economical utilization and sustain-

able development of water resources can provide a guarantee for the high-quality development of the Yellow River Basin.

**Author Contributions:** Conceptualization, X.G.; methodology, X.G.; software, Q.D.; validation, Q.D. and B.W.; formal analysis, B.W.; investigation, W.Z.; resources, W.Z.; data curation, W.Z.; writing—original draft preparation, Q.D.; writing—review and editing, Q.D., B.W. and W.Z. All authors have read and agreed to the published version of the manuscript.

**Funding:** This research was funded by the National Natural Science Foundation of China (No. NSCF-51979119) and by the Basic R & D Special Fund of Central Government for Non-profit Research Institutes (No. HKY-JBYW-2020-17).

**Institutional Review Board Statement:** It is not applicable for this study not involving humans or animals.

**Informed Consent Statement:** It is not applicable for this study not involving humans.

**Data Availability Statement:** The data that support the findings of this study are available in this article.

**Acknowledgments:** The authors would like to thank the research team for their help.

**Conflicts of Interest:** The authors declare no conflict of interest.

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
