# Peer review of "Study on Water Rights Allocation of Irrigation Water Users in Irrigation Districts of the Yellow River Basin"

_water, doi:10.3390/w13243538_

Round 1

Reviewer 1 Report

Comments to the article:

The article tries to address the important issue of study on water allocation of irrigation water users, this issue by all means is very interesting in the aspect of climate change and very serious problems of water shortage for agricultural purposes.

In the abstract there is only one result from the research carried out in this article;

The introduction includes a literature review with only 15 publications on this topic of which 13/15 are literature items from 1 country

Materials and methods: line 92-93 how to explain such low rainfall values 144.5 mm and huge evaporation values (2377.1 mm) where does the water for this process come from ?

Lack of basic characteristics of the 411 irrigation canals (length, slopes, depth, slope inclination, soil in which they are located, groundwater table position under canals)

Figures 1 and 2 not very clear, especially figure 2 nothing is clear from it

Formulas 1-4: no value and range of coefficients in these formulas, similarly in the following formulas (11-12) (13-14), it is difficult to follow the results of calculation

Fig. 3 invert the vertical axis in the figure

The research procedure does not arouse objections, nevertheless the final results are difficult to verify without knowing the values of particular coefficients adopted earlier

The title of the paper and its topic is interesting but it still needs a lot of work. At its current level I do not recommend it for further procedure in the journal MDPI Water. At its current level it can be, after corrections, applied to a journal of local (national) scale. I did not review the article with a bibliography of only 15 literature items, this requires a definite extension of the state of knowledge not only of the authors from 1 country.

Author Response

Reviewer #1: The article tries to address the important issue of study on water allocation of irrigation water users, this issue by all means is very interesting in the aspect of climate change and very serious problems of water shortage for agricultural purposes.

Response: Thank you very much for the reviewer's recognition of our research direction.

1.In the abstract there is only one result from the research carried out in this article;

Response: Thanks for your comment. We have added the results of this article to the abstract: Results show that the allocated water rights of the national canals in the irrigation district are less than the current, for example, water rights of the Grazing team (4) canal are reduced by 7.3 ten thousand m³ than before, in which water rights of farmer household 1, 2, 3 and 4 get compensation and 5, 6, 7 and 8 are cut by the water rights allocation model and the Gini coefficient is reduced from 0.1968 to 0.1289. Please see L18-25 of the revised manuscript.

2.The introduction includes a literature review with only 15 publications on this topic of which 13/15 are literature items from 1 country;

Response: Thanks for your comment. We fully agree with you. The introduction of the article has been greatly deleted, which has reduced the references of our country, increased the references of other countries, and expanded the knowledge state of this article. At present, a total of 24 references have been expanded. Please see the revised manuscript. At the same time, your valuable suggestions have also played a good role in guiding and helping us write other papers in the future.

3. Materials and methods: line 92-93 how to explain such low rainfall values 144.5 mm and huge evaporation values (2377.1 mm) where does the water for this process come from?

Response: Thanks for your comment. I'm sorry we didn't explain it clearly in the article, because of the small precipitation, the water resources of Wulanbuhe Irrigation Area are very scarce. In order to meet the local water demand, it is necessary to use the transit Yellow River water, which has a certain water intake index for this area. Therefore, the evaporation value is large, and the water in the evaporation process mainly comes from the water consumption of the Yellow River. We have added notes in the manuscript. Please see L102-104 of the revised manuscript.

4. Lack of basic characteristics of the 411 irrigation canals (length, slopes, depth, slope inclination, soil in which they are located, groundwater table position under canals)

Response: Thanks for your comment. As you said, 411 irrigation canals have many basic characteristics (length, slopes, depth, slope inclination, soil in which they are located, groundwater table position under canals). Different basic characteristics need to be used for different research purposes. The purpose of this paper is to study the water rights allocated under the multi-year average of 411 irrigation canals. The basic feature of 411 irrigation canals is the average water diversion in recent 5 years. However, your valuable suggestions have broadened our team's future research vision. Thank you very much.

5. Figures 1 and 2 not very clear, especially figure 2 nothing is clear from it

Response: Thanks for your comment. We have redrawn figures 1 and 2 to improve their clarity. Please see L110 and L129 of the revised manuscript.

6. Formulas 1-4: no value and range of coefficients in these formulas, similarly in the following formulas (11-12) (13-14), it is difficult to follow the results of calculation

Response: Thanks for your comment. I'm sorry we didn't explain it clearly in the article, we have explained the value and range of the coefficients of formulas (1-4) and formulas (11-14) in the article. Please see L147-154, L187-190, L199-202 and L323-324 of the revised manuscript.

7. Fig.3 invert the vertical axis in the figure

Response: Thanks for your comment. We have reversed the vertical axis of the graph. Please see L230 of the revised manuscript.

8. The research procedure does not arouse objections, nevertheless the final results are difficult to verify without knowing the values of particular coefficients adopted earlier

Response: Thanks for your comment. ηi、ηi'  are respectively the canal system water utilization coefficient before and after the lining of the canal i ; ( 0<ηi<ηi'<1)). The reduction ratio s depends on the regional attention to the principle of equity. We consulted the local water resources management department and took the specific coefficient of 0.3. The results can be reflected from the change of Gini coefficient value of objective function. Before balanced distribution, the Gini coefficient is 0.1968. After balanced distribution, the Gini coefficient is adjusted to 0.1289.

9. The title of the paper and its topic is interesting but it still needs a lot of work. At its current level I do not recommend it for further procedure in the journal MDPI Water. At its current level it can be, after corrections, applied to a journal of local (national) scale. I did not review the article with a bibliography of only 15 literature items, this requires a definite extension of the state of knowledge not only of the authors from 1 country.

Response: Thanks for your comment. We are glad that you are interested in our research direction. According to your comments, our team has conducted in-depth discussion and modification of the article. We have deleted some references of authors in our countries / regions and added references in other countries / regions. In addition, we have also made a lot of modifications to other contents of the article, improved the research content of the article and corrected the details of chart symbols. I hope my article can be published in your journal.

Reviewer 2 Report

Excellent concept presented very well.  I can't wait to try this out in my region!

Author Response

Reviewer #2: Excellent concept presented very well.  I can't wait to try this out in my region!

Response: We sincerely appreciate your high recognition of our research. Our team will carry out further research in this research direction.

Reviewer 3 Report

This work evaluates the water right allocation of irrigation water users in yellow river basin. The authors showed that it is important to study serious problems of water security and lack of water for households and farmers in the Yellow river basin. The topic fits the scope of Water, and the analysis is adequate. However, I find some descriptions are unclear, and the results can be better presented. Some suggestions are as below.

  1. Adjust keywords - some of them are the same in the paper title
  2. The aims of study highlight in the introduction (e.g. number)
  3. The section of Material and methods is very long, I suggest to shorten it and more highlight particular parts (to make it more clear to readers)
  4. I am missing in the section of Discussion some papers which deal with this kind of topic - please add at least 2-3 papers
  5. Please add to Conclusions some results from your study.

Author Response

Reviewer #3: This work evaluates the water right allocation of irrigation water users in yellow river basin. The authors showed that it is important to study serious problems of water security and lack of water for households and farmers in the Yellow river basin. The topic fits the scope of Water, and the analysis is adequate. However, I find some descriptions are unclear, and the results can be better presented. Some suggestions are as below.

1.Adjust keywords - some of them are the same in the paper title

Response: Thanks for your comment. We have changed some keywords of the article, we have deleted the keywords “Irrigation district”, “Water right allocation”, and added Water-saving potential. Please see L26 of the revised manuscript.

2. The aims of study highlight in the introduction (e.g. number)

Response: Thanks for your comment. We have made a lot of modifications to the introduction of the article, emphasizing the research purpose of the article through figures and other forms. For example, water rights of the Grazing team (4) canal are reduced by 7.3 ten thousand m³ than before, in which water rights of farmer household 1, 2, 3 and 4 get compensation and 5, 6, 7 and 8 are cut by the water rights allocation model and the Gini coefficient is reduced from 0.1968 to 0.1289. Please see L18-25 of the revised manuscript.

3. The section of Material and methods is very long, I suggest to shorten it and more highlight particular parts (to make it more clear to readers)

Response: Thanks for your comment. Thank you for your valuable comments. We have properly simplified the materials and methods of the article, highlighting the research focus of the article. Please see the revised manuscript.

4. I am missing in the section of Discussion some papers which deal with this kind of topic - please add at least 2-3 papers

Response: Thanks for your comment. We fully agree with you, and we have added 3 papers which deal with this kind of topic in the discussion part of the article Please see the revised manuscript.

5. Please add to Conclusions some results from your study.

Response: Thanks for your comment. We have added the results of this article to the conclusion of the article. Please see the revised manuscript.

Round 2

Reviewer 1 Report

In this form I would like to suggest to publish this material in MDPI Water journal , most of my suggestion were taken into account

Reviewer 3 Report

Authors accepted all recommendations, I recommend to publish this paper.